# Examining Social Equity in the Co-Management of Terrestrial Protected Areas: Perceived Fairness of Local Communities in Giant Panda National Park, China

**Qiujin Chen [1], Yuqi Zhang [2] , Yin Zhang [1],* and Mingliang Kong [1]**

[1] School of Architecture and Urban Planning, Chongqing University, Chongqing 400030, China
[2] Department of Cultural Geography, Faculty of Spatial Sciences, University of Groningen, 9700 AV Groningen, The Netherlands
* Correspondence: yinzhang@cqu.edu.cn

**Abstract:** Social equity is imperative both morally and instrumentally in the governance of protected areas, as neglecting this consideration can result in feelings of injustice and thus jeopardize conservation objectives. Despite the progressive attention paid to conservation equity, few have linked it with co-management arrangements, especially in the context of terrestrial protected areas. This study assesses the fairness perceptions in China's Giant Panda National Park from recognitional, procedural, and distributional dimensions, to further disclose their correlations with individuals' characteristics and participation in co-management activities. The regression analysis shows that all co-management types (instruction, consultation, agreement, and cooperation) are significantly linked with certain directions of perceived social equity. One novel finding here is that alternative types of co-management activities are influencing social equity in different ways. In addition, our research discloses the effects of education across all equity categories, and location is merely significantly related to recognitional equity. These findings suggest more inclusive and empowered co-management endeavors to strive for more equitably managed protected areas. Crucial steps to advance this include extending participative channels, co-producing better compensation plans, strengthening locals' conservation capabilities, etc. Herein, this study appeals to a greater focus on social equity issues in co-management regimes, and tailored actions should be taken to tackle specific local problems.

**Keywords:** protected areas; co-management; social equity; fairness perception; empowerment levels

## 1. Introduction

Protected areas are essential not only to sustain biodiversity and ecosystem services, but also to support local livelihood and well-being [1]. By no means should indigenous people and local residents be forced into victims and refugees of the global expansion of protected areas [2]. Over the last two decades, there have been concerted efforts globally to make protected areas more effectively and equitably managed, mostly for the benefits of local communities [3,4]. The slogan of "equity and benefit sharing" was put forward by the Convention on Biological Diversity's Programme on Protected Areas in 2004. Furthermore, the principle that protected areas should be "effectively and equitably managed" was highlighted by the Aichi Biodiversity Target 11 in 2010 [5], which was later strengthened by International Union for Conservation of Nature (IUCN) World Parks Congress held in 2014 [6]. The better understanding and consideration of social equity issues in protected areas are believed to deliver better conservation outcomes, as protected areas can seldomly survive without strong and firm social support from their surroundings [7–9].

The recent 5 years have witnessed a considerable increase in the number of studies focusing on the social equity aspects of protected areas [10–12]. Zafra-Calvo et al. (2017) established an indicator system to assess the equitable management of protected areas from

recognitional, procedural, and distributional dimensions, and later applied this framework to evaluate their interrelations among 225 protected areas globally [13,14]. Bennett et al. (2020) expanded and enriched those indicators to capture the fairness perceptions of small-scale fishermen in marine protected areas [15]. While some authors were inclined to look at social equity issues from the perspective of distribution [16,17], others paid more attention to the procedural or recognitional dimension [18,19].

Among all those researches, very few have linked social equity with co-management of protected areas. Despite the fact that there is no commonly accepted concept for co-management, this term is most frequently comprehended as the sharing of rights and responsibilities among the governments, local resource users, and other partners (Carlsson and Berkes 2005; Borrini-Feyerabend 2007) [20,21]. In addition, the majority of current studies in this aspect are set in the context of marine protected areas or fisheries [16,22,23], not in terrestrial protected areas, the co-management of which also displays significant roles in forest, grassland, and biodiversity conservation [24,25]. Although several studies have demonstrated how demographic attributes and social-economic characteristics, such as gender, education level, and household wealth, can have impact on fairness percep-tions of local communities toward the co-managed marine protected areas, none of these have considered the influence of their involved co-management types [15,16]. Due to the complexity and plurality of co-management mechanisms, local stakeholders are usually in-volved in different co-management types and forms, showing the variability in perceptions, attitudes, and behaviors toward protected areas, which are frequently related to social equity issues [26,27]. To understand the correlation between participative co-management activities and the fairness perceptions of grassroots is vital to achieve better social outcomes of co-management in protected areas. On the contrary, the lack of this consideration in enforcing co-management programs in protected areas can result in serious social conflicts, and consequently lead to poor conservation performance [28].

In this paper, we aim to explore how participative co-management activities can have influences on locals' fairness perceptions in a newly designated terrestrial protected area in southwestern China. Our research hypotheses are listed as follows: (1) Individuals' demographic characteristics (e.g., gender, age, residency year, education, and profession) can have influences on their fairness perceptions; (2) some household features (e.g., villages, household size, migrant workers, annul income, and income sources) are associated with individuals' perceived fairness; (3) the number and type of participative co-management activities are positively linked with villagers' fairness perceptions. In the IUCN guideline of good governance of protected areas, those co-management arrangements diversified into five types, namely, instructive, consultative, agreement, cooperation, and empow-erment, based on their empowerment levels [29]. Moreover, in this study, we classify diverse co-management activities in Giant Panda National Park according to the IUCN classification. With respect to the measurement of fairness perception, we largely borrow from Zafra et al. (2017) [13] and Bennett et al. (2020) [15], while making minor adjustments according to the study site. Both quantitative and qualitative methods were adopted herein. Through this explanatory study, we seek to disclose the relations between participated co-management types and fairness perceptions of locals from the recognitional, procedural, and distributional perspectives, contributing to the empirical evidence on the social equity of co-managed terrestrial protected areas, and producing practical and theoretical insights for the co-management policy and practice in protected areas.

## 2. Materials and Methods

### 2.1. Study Site

The Giant Panda National Park (GPNP) is located in the southwest China as part of the Minshan and Qionglai Mountains, covering a total area of roughly 21,978 km$^2$. It was first promoted as a national park pilot in 2016 and then officially recognized as one of China's first batch of national parks in 2021 [30]. The GPNP is integrated and expanded from 73 existing protected areas, and is further divided into 4 regions after being designated as a

national park. With a total area of approximately 400 km², the Tangjiahe area is situated in the northeast region of GPNP in Sichuan Province, with the protection of giant pandas and their habitats as the primary conservation objectives (Figure 1). In addition, this area has been assigned as a nature reserve since 1978, with over a 40-year history of conservation.

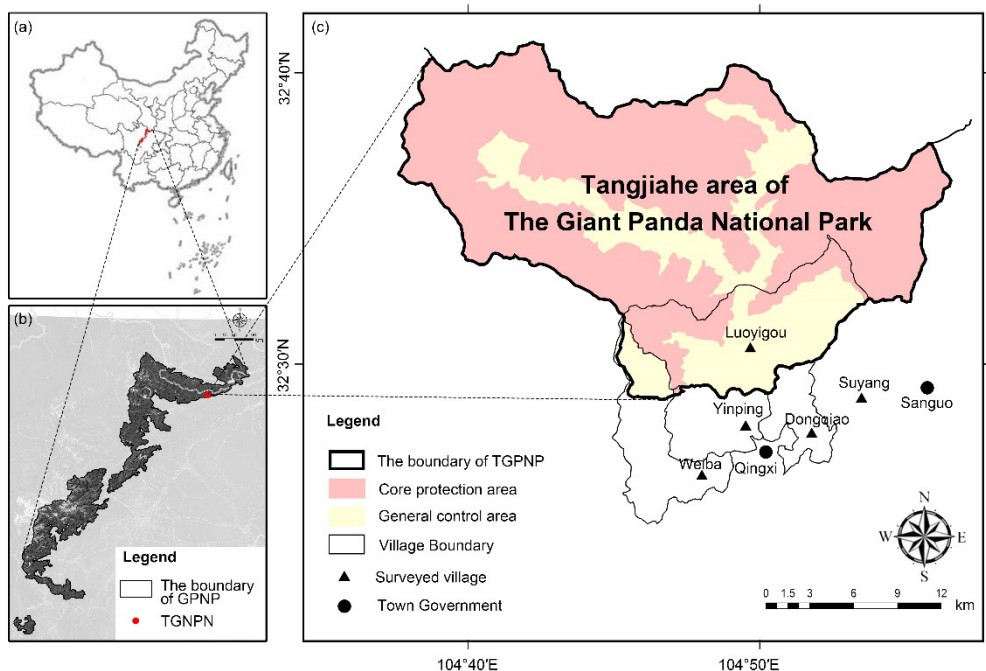

**Figure 1.** Map of the Tangjiahe area of the Giant Panda National Park: (**a**) Location of GPNP in China; (**b**) location of TGPNP; (**c**) location map of the TGPNP and surveyed villages. (Note: i. The figure does not include the spatial boundary of Suyang village as this information is not available for our research group. ii. The location of surveyed villages is positioned at the office of the village committee).

Tangjiahe area of GPNP (TGPNP) is selected as our study site to explore the fairness perception of community-based co-management in protected areas for the following two reasons. First, the Administration of Tangjiahe Area (ATA) has started to enforce community-based efforts (e.g., joint fire prevention and infrastructure building supports) with its surrounding communities since 1978, and has tried various co-management strategies, such as organizing co-management committees, signing co-management agreements, and arranging industrial guidance, as well as introducing foreign and domestic NGOs to develop a differentiated co-management model in surrounding villages. Those co-management arrangements appear to be super abundant and diverse until now, yet the social effects remain to be uncovered [31]. Second, as establishing community-based co-management mechanisms was put forward as one of the critical strategies in the construction of China's national park system in 2017, the GPNP positively responded to the summoning of the central government and largely facilitated community-based tourism [32]. The locals' fairness perceptions toward those co-management countermeasures are essential to be disclosed, as it might affect the conservation outcomes and performances.

There are 7 villages bordering TGPNP in Qingxi and Sanguo Town, with a population of about 9500. To identify the specific villages suitable for our in-depth survey, we consulted with officials working at the ATA, and two criteria were adopted after repeated discussion. First, there are stable and long-lasting co-management arrangements settled between villages and the ATA. Second, those co-management models need to be both representative and differentiated. By this method, villages of Yinping, Luoyigou, Weiba, Dongqiao, and Suyang were selected as a result, and the basic information was listed in Table 1. Among those villages, Luoyigou village is the only one located in the General Control Area within the boundary of the TGPNP, with ecological restoration and habitat enhancement as its

main conservation objective. Due to this reason, the village has suffered from severe human-wildlife conflicts for years. Therefore, ATA has established a co-management committee since 2018 and a human-wildlife compensation program was specifically launched in 2019 in Luoyigou. In addition, as the gateway community of Tangjiahe area, Yinping village has been greatly supported financially and technically by the ATA and the Qingchuan County Government since 1997 to promote tourism development. More specifically, an agricultural cooperative was established in Suyang village to facilitate local development. Apart from the aforementioned arrangements, forest ranger programs and infrastructure construction were enforced by the ATA in all five villages, while beekeeping training was organized in four villages except for Suyang.

**Table 1.** Basic information of the five selected villages.

| Villages | Population Size | Area | Main Industries | Key Co-Management Strategies |
|---|---|---|---|---|
| Luoyigou | 1085 | 62 km$^2$ | Tourism, agriculture, and cultivation | Co-management committee, human-wildlife conflicts compensation, tourism support, beekeeping training, infrastructure construction, and forest rangers |
| Yinping | 1823 | 39.7 km$^2$ | Tourism, agriculture, and cultivation | Co-management committee, tourism support, beekeeping training, infrastructure construction, and forest rangers |
| Weiba | 870 | 66.12 km$^2$ | Tourism, agriculture, stone production | Beekeeping training and forest rangers |
| Dongqiao | 1160 | 27.92 km$^2$ | Agriculture, cultivation, and tourism | Beekeeping training and forest rangers |
| Suyang | 1389 | 22 km$^2$ | Agriculture and cultivation | Establishing an agricultural cooperative |

*2.2. Survey Sampling Methods and Design*

A pre-survey field was conducted in July 2017 to interview ATA staff to collect basic information about co-management arrangements of TGPNP, and both informal discussions with local people and formal interviews with ATA staff were conducted to select the most suitable criteria to assess co-management activities and the perceived fairness of locals. Questionnaires were distributed on-site from 29 June 2022 to 7 July 2022 in TGPNP. This survey took the household as the basic unit and selected one person with the most frequent contacts with the ATA, recommended by household members. Random sampling was used to select the respondents, and the specific number of respondents was determined according to the population size of the village. In this way, a total of 428 questionnaires were collected by the research team. After excluding 4 invalid questionnaires, the respondents of which came from non-survey villages, the actual valid samples reached 424, with a 99.3% effective return rate.

The questionnaire consists of four sections. The first two sections include a broad set of questions related to the demographics (e.g., gender, age, education, location, occupation) of local residents and their household characteristics (e.g., household income, household size, income sources, and residency year), as well as the co-management activities they are involved in. Those 12 types of co-management activities are inducted from a total of 15 co-management arrangements after the discussion with ATA staff (see Supplementary Materials—Table S4). Those activities are classified into four categories based on an increasing level of empowerment of local communities. While instructive co-management refers to those community-based measures where the ATA takes the lead and communities

simply follow the instructions, consultative co-management means better information exchange between both sides. The responsibilities and benefits of conservation are clearly and formally divided among different stakeholders in the agreement type of co-management. Furthermore, cooperation is the co-management typology where the participants can partially be delegated in the decision-making or enforcement of conservation affairs, which is the highest empowerment level recognized in TGPNP. All those activities are assessed by Yes ("participated") or No ("not participated"), listed in Table 2.

**Table 2.** Co-management types and activities.

| Category | Activity Number | Co-Management Activities |
|---|---|---|
| Instruction | A1 | Energy transformation and other infrastructure building projects |
| | A2 | Skill training and industrial support activities |
| | A3 | Environmental educational activities |
| Consultation | A4 | Community-based co-management meetings |
| | A5 | Consultative meetings for planning and policy making |
| | A6 | Easy access to co-management Information |
| Agreement | A7 | Agreements of fire prevention and human-wildlife conflict compensation |
| | A8 | Agreements of community-based co-management |
| | A9 | Benefits sharing of bee farming and other cooperatives |
| Cooperation | A10 | Fire prevention and forest patrolling work |
| | A11 | Participation in enacting conservation rules |
| | A12 | Accountability for some conservation affairs |

The last section of the questionnaire is concerned with locals' perceptions of fairness toward TGPNP, measured through statements developed for each dimension of social equity. In this section, we borrowed from Zafra-Calvo et al. (2017) [13], Lou Lecuyer (2019) [33], Nathan J. Bennett et al. (2020) [15], and Georgina G. Gurney (2021) [16], while making minor adjustments and adding additional attributes according to our study site. For recognitional equity, an item concerning land ownership was added as land conflicts were frequently recognized by interviews. For procedural equity, we deleted the indicator of access to justice, since no conflict resolution mechanisms were found in TGPNP. From the distributional perspective, two attributions of wildlife compensation and empowerment distribution were added for the wildlife-human conflict compensation and the forest ranger programs launched in TGPNP. All those questions are measured in a 5-point Likert scale in this section, listed in Table 3.

**Table 3.** Selected indicators to measure social equity in GPNP.

| Category | Attribute | Survey Questions |
|---|---|---|
| Recognition | culture | GPNP respects our local culture and traditional customs |
| | livelihood | GPNP imposes no negative impact on my original livelihood |
| | Legal and traditional rights | GPNP can sincerely respect my legal and traditional rights |
| | Land ownership | I declare no land ownership conflicts with GPNP |
| | Traditional knowledge | Traditional knowledge can be effectively involved in the management of GPNP |

**Table 3.** *Cont.*

| Category | Attribute | Survey Questions |
|---|---|---|
| Procedure | Decision making | I can fully express my opinion and effectively be involved in the decision-making process of GPNP |
| | Participation | GPNP has convenient channels and fair procedures to encourage local participation |
| | transparency | The information of conservation decisions and reasons for decisions are readily available |
| | Accountability | I understand the responsibility of ATA and know to whom to raise concerns to solve issues related to management actions |
| | Free, prior, and informed consent (FPIC) | When ATA issues plans and policies addressed to me, I will be informed in advance |
| Distribution | Conservation burdens | I fairly bear the responsibility of conservation in GPNP, compared to other local residents |
| | Ecological compensation | I am satisfied with the ecological compensation made by GPNP |
| | Wildlife conflicts compensation | I can easily get appropriate compensation from human-wildlife conflicts |
| | Benefits distribution | I can fairly get economic benefits from co-management, compared to other local residents |
| | Employment distribution | I can fairly get employment opportunities from ATA, compared to others |

Qualitative methods are also used as a supplementary approach in this research. Seventeen semi-structured interviews were conducted with different stakeholders: Staff in the Community Office of the ATA, officials of Qingxi and Sanguo town governments, as well as village leaders and elites. The selection of stakeholders is based on the correlation to co-management, such as people with rights, with official information, and prestigious local people, as well as considerations of the equilibrium of gender and age. The purpose of these interviews is to identify the equitable issues of TGPNP and select the most suitable criteria to assess co-management activities and fairness perceptions. In addition, secondary data (e.g., research reports, government reports and plans, and statistics) were collected and analyzed to understand the contexts.

*2.3. Data Analysis*

All data analysis was completed in SPSS 26.0 (IBM Corp., Armonk, NY, USA). First, the reliability analysis was performed in this study using Cronbach's alpha index. In this study, the overall Cronbach's alpha coefficient was 0.831, which is above the eligible index of 0.7, indicating that the obtained survey results had good internal reliability. The content validity was also assessed here. The figures for all items were all significantly correlated at the 0.01 level, signifying positive outcomes in content validity.

First, we calculated the score for each equity dimension (recognitional, procedural, and distributional equity) by the average score of five indicators in this category, and then built the score for combined equity by the mean score of all 15 indicators. Second, we tested for univariate associations (one-way ANOVA and Spearman correlation analysis) between recognitional, procedural, distributional, and combined equity scores and the demographic characteristic and participative factors. While one-way ANOVA was used for categorical variables (e.g., gender, occupation, and villages), Spearman correlation analysis was utilized for ordinal variables, such as education level, household size, and income, as well as the number of participative co-management arrangements. Finally, linear regression analysis was adopted here to develop regression models for each composite social equity score using variables (e.g., age, education, annual household income) significantly correlated to equity perception, to further disclose their intertwined relations.

### 2.4. Sample Description

Our sample consisted of 424 residents who lived within or surrounding TGPNP (Table 4), with 46.0% male and 54.0% female. The majority of respondents were in older age brackets, with 72.6% (*n* = 308) older than 50. Their education levels were generally low, since most respondents (65.1%) had only completed primary or junior school and even 22.4% had never attended any school. In addition, the vast majority of respondents lived here for more than 20 years (88.3%) and made a living by farming (76.4%).

**Table 4.** Description of respondents involved in this survey.

| Survey Item | Category | Frequency (*n* = 424) | Percentage (%) |
|---|---|---|---|
| Gender | Male | 195 | 46.0 |
| | Female | 229 | 54.0 |
| Age | Under 40 | 56 | 13.2 |
| | 41–50 | 60 | 14.2 |
| | 51–60 | 135 | 31.8 |
| | 61–70 | 94 | 22.2 |
| | Over 70 | 79 | 18.6 |
| Education | No school | 324 | 22.4 |
| | Primary school | 23 | 41.5 |
| | Junior school | 36 | 23.6 |
| | High school | 41 | 8.5 |
| | Undergraduate and above | 95 | 4.0 |
| Residency years | Under 10 | 176 | 4.2 |
| | 10–20 | 100 | 7.5 |
| | Over 20 | 36 | 88.3 |
| Professional | Farmers | 17 | 76.4 |
| | Employees | 104 | 5.4 |
| | Merchants | 139 | 8.5 |
| | Other | 73 | 9.7 |
| Villages | Luoyigou | 60 | 24.5 |
| | Yinping | 48 | 32.8 |
| | Weiba | 18 | 17.2 |
| | Dongqiao | 32 | 14.2 |
| | Suyang | 374 | 11.3 |
| Household size | 1–3 | 115 | 27.1 |
| | 4–6 | 261 | 61.6 |
| | 7–9 | 41 | 9.7 |
| | >10 | 7 | 1.7 |
| Household migrant workers | 0 | 144 | 34.0 |
| | 1 | 125 | 29.5 |
| | 2 | 99 | 23.3 |
| | >3 | 56 | 13.2 |
| Annual household income (RMB) | Less than 10,000 | 153 | 36.1 |
| | 10,001–30,000 | 131 | 30.9 |
| | 30,001–60,000 | 78 | 18.4 |
| | 60,001–100,000 | 37 | 8.7 |
| | More than 10,001 | 25 | 5.9 |
| Household source of income | Farming | 195 | 46.0 |
| | Tourism | 71 | 16.7 |
| | Forestry | 18 | 4.2 |
| | Local employment | 106 | 25.0 |
| | Nonlocal employment | 136 | 32.1 |
| | Other | 68 | 16.0 |

The average household size of those respondents was five people, and most of the households had none or only one migrant worker, accounting for 34% and 29.6%, respectively. Most of the surveyed households had a relatively low annual household income, with 36.4% earning less than RMB 10,000 and 30.9% earning between RMB 10,001 and 30,000 per year. Despite the fact that the main household income sources were farming and

non-local employment, there were also households that made a living by local employment, tourism, and forestry, the percentages of which were 25.0%, 16.7%, and 4.2%, respectively.

## 3. Results

### 3.1. Fairness Perceptions toward TGPNP

The descriptive analysis showed that perceptions of recognitional equity were more positive (Mean = 3.59), compared to those of procedural equity and distributional equity (Mean = 2.60 and 2.81, respectively). As shown in Figure 2, indicators related to recognitional equity were heavily skewed toward positive judgements, indicating that recognitional equity was most likely to be perceived as fair. Significantly, most respondents (67%, 70.1%, 56.8%, and 78.8% respectively) "strongly agreed" or "agreed" to the four indicators of culture, livelihood, legal and traditional rights, and land ownership. By contrast, all indicators related to procedural fairness were strongly skewed toward negative perceptions, especially regarding community participation, where most respondents (66.3%) felt they were not truly involved in the planning and management of TGPNP. Similarly, 47.1% of the respondents "disagreed" or "strongly disagreed" about effective decision-making. In addition, indicators of distributional equity showed dissimilar results. The data showed that 67.9% of respondents believed they were equally responsible for forest fire prevention, while merely one-fifth of respondents "agreed" or "strongly agreed" with the appropriate amount of ecological compensation and wildlife conflict compensation. Moreover, perceptions of distribution of benefits and employment were balanced between positive and negative, since a considerable proportion of respondents (41.7% and 33.5%, respectively) did not have access to the relevant information.

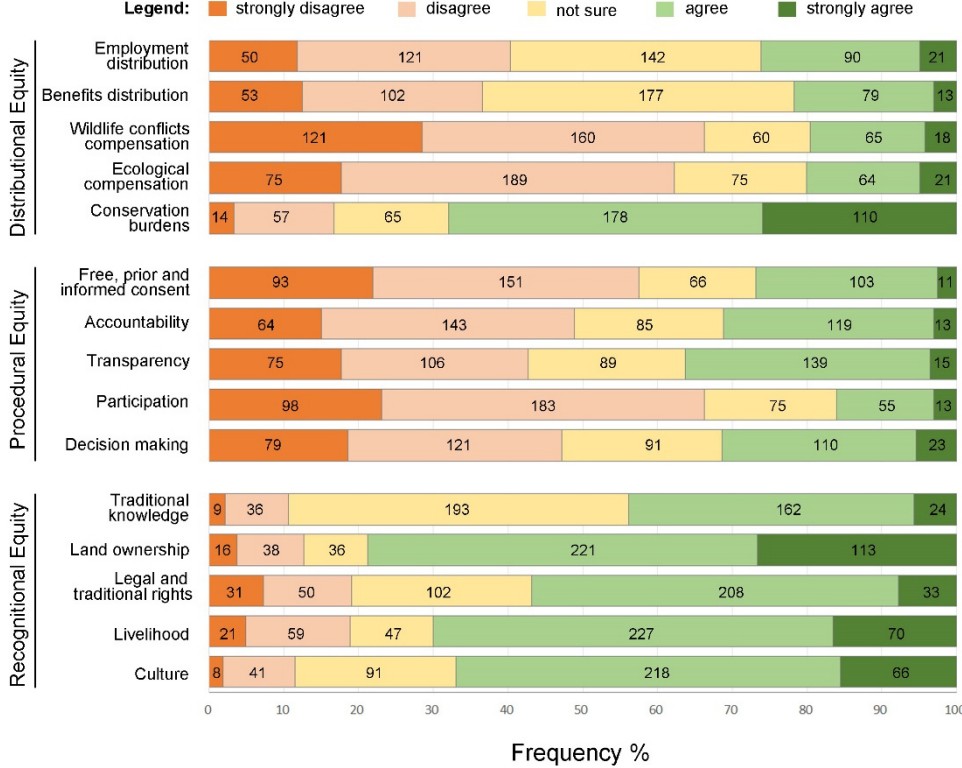

**Figure 2.** Stacked bar charts showing frequency distributions for all individual social equity indicators. The numbers within the bars indicated the number of respondents (further details are provided in Supplementary Materials—Table S7).

### 3.2. Participative Co-Management Activities

Descriptive analysis showed varied participation rates among different co-management activities (see Figure 3 and Supplementary Materials—Table S5). The most frequently participated co-management activity was energy renovation arrangements (A1 = 74.5%), followed by environmental education activities (A3 = 56.4%). The percentages for the remaining 10 co-management activities were all below 40%, with co-management agreements and conservation accountability ranking the lowest two (A8 = 9.2%, A12 = 5.4%). Furthermore, we contrasted the numbers of participants across four empowering levels of co-management, with the instruction type being the largest, followed by those of consultation and agreement, and finally, the cooperation type. It was clear that the number of participants tended to decline with the increase in co-management empowering levels. Statistics also showed that the majority of respondents (*n* = 273, 64,4%) were involved in less than three co-management events. Notably, 4.5% (*n* = 19) of respondents had none of this experience (see Figure 4 and Supplementary Materials—Table S6).

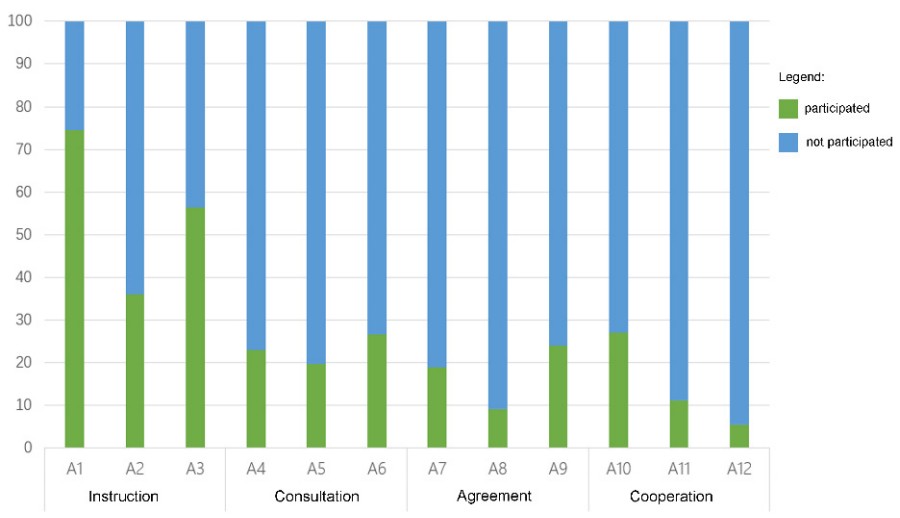

**Figure 3.** Participation frequency in diversified co-management activities. The horizontal coordinates represent different co-management activities, and the vertical coordinates represent the percentages of participants (*n* = 424).

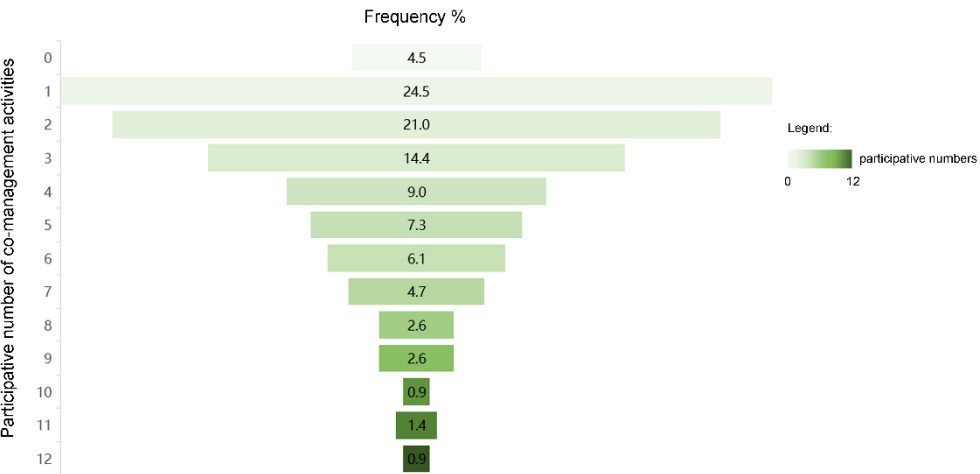

**Figure 4.** Frequency of the number of co-management activities in which respondents participated. The numbers on the bar chart represent the percentage of respondents for a certain participated number (*n* = 424).

### 3.3. Analysis of Correlation

Results of the Spearman correlation analysis showed that all types of co-management were significantly correlated with each dimension of social equity, listed in Table 5. When it came to the number of participated co-management activities, the results were similar. Those findings indicated that the more local residents were involved in co-management arrangements, the more likely they would have positive feelings for recognitional, procedural, distributional, and composite fairness.

**Table 5.** Summary of results from a univariate model of the relationship between predictors (types and numbers of co-management activities) and social equity perceptions. (Note: The data in the table showed correlation coefficients. Significance levels: * = $p < 0.05$, ** = $p < 0.01$).

| Category | Recognitional Equity | Procedural Equity | Distributional Equity | Combined Social Equity |
|---|---|---|---|---|
| Instruction | 0.190 ** | 0.278 ** | 0.321 ** | 0.333 ** |
| Consultation | 0.177 ** | 0.389 ** | 0.325 ** | 0.399 ** |
| Agreement | 0.102 * | 0.216 ** | 0.279 ** | 0.250 ** |
| Cooperation | 0.169 ** | 0.374 ** | 0.392 ** | 0.411 ** |
| Number of participated co-management activities | 0.198 ** | 0.373 ** | 0.400 ** | 0.418 ** |

With respect to the socio-demographic features, one-way ANOVA and Spearman correlation analysis were adopted accordingly. Spearman correlation analysis revealed that education and annual household income were significantly and positively correlated with all directions of fairness perceptions, while age was negatively related to all. Results of the one-way ANOVA test showed that villages were merely significantly related to recognitional equity. Additionally, two household sources of income, tourism, and forestry, were recognized as significantly correlated factors for certain dimensions of fairness perceptions (Table 6).

**Table 6.** Summary of results from a univariate model of the relationship between predictors and social equity perceptions. (Note: The symbols + or − indicate the direction of the relationship between fixed factors and ordinal levels. +: Positive correlation, −: Negative correlation. Significance levels: / = Not significant, * = $p < 0.05$, ** = $p < 0.01$).

| Category | | Analysis Method | Recognitional Equity | Procedural Equity | Distributional Equity | Combined Social Equity |
|---|---|---|---|---|---|---|
| Gender | | One-way ANOVA | / | / | / | / |
| Age | | Spearman | − * | − ** | − ** | − ** |
| Occupation | | One-way ANOVA | / | / | / | / |
| Education | | Spearman | + ** | + ** | + ** | + ** |
| Villages | | One-way ANOVA | ** | / | / | / |
| Residency years | | Spearman | / | / | / | / |
| Household size | | Spearman | + ** | / | / | / |
| Household migrant workers | | Spearman | / | / | / | / |
| Annual household income | | Spearman | + ** | + * | + ** | + ** |
| Household source of income | Farming | One-way ANOVA | / | / | / | / |
| | Tourism | One-way ANOVA | / | / | ** | ** |
| | Forestry | One-way ANOVA | / | ** | ** | ** |
| | Local employment | One-way ANOVA | / | / | / | / |
| | Non-local employment | One-way ANOVA | / | / | / | / |

*3.4. Regression Equation*

We conducted the linear regression analysis to assess how social demographics and participative factors could have influence across all directions of perceived fairness (see Supplementary Materials—Model 1–4). The first regression model (Adjusted $R^2$ = 0.107, F = 6.653, *p*: <0.0001) clearly showed the impact of education, village, household size, and consultation on recognitional equity. Among them, household size has the largest effect, followed by education and consultation, with village being the smallest one. The second model (Adjusted $R^2$ = 0.272, F = 20.756, *p*: <0.0001) showed the influence from education, consultation, and cooperation on perceived procedural equity. In this model, cooperation has the largest effect on procedural equity, while education was the smallest. The third model (Adjusted $R^2$ = 0.27, F = 18.423, *p*: <0.0001) disclosed the causal relationship between the four types of co-management and perceptions of distributional equity, among which the cooperation continued to have the largest effect.

In addition, a linear regression analysis was conducted to assess the effects of those indicators on perceived combined equity, marked as model four (Adjusted $R^2$ = 0.321, F = 23.26, *p*: <0.0001). The significant impact was detected from variables of education, construction, consultation, and cooperation, among which the cooperation type of co-management has the largest impact. Above all, the aforementioned four models all passed collinearity diagnosis, serial correlation diagnosis, and residual normality test, thus partially reflecting the causal relationship between the relevant variables and to some extent assisting us in better understanding their influences on fairness perceptions.

## 4. Discussion

*4.1. Relationships between Perceived Fairness and Socio-Demographic Characteristics*

Our findings disclose that locals' fairness perceptions are significantly associated with several socio-demographic characteristics, including education, location, and household size. Among them, the level of education is the most widely related factor, concerning not only recognitional (Beta = 0.164, *p* = 0.004) and procedural equity (Beta = 0.165, *p* = 0.002), but also combined equity (Beta = 0.167, *p* = 0.001). This suggests that higher education levels of local residents generally lead to better exposure to CBCM information and participative opportunities, and this can consequently link to a better understanding and acceptance of conservation justice. This result aligns with researches conducted by Nathan J. Bennett et al. (2020) [15], Lou Lecuyer et al. (2019) [33], and Aires Afonso Mbanze et al. (2021) [34], demonstrating the impact of formal education on perceived fairness of conservation. By contrast, Georgia G. Gurney et al. (2021) [16] discloses the association between formal education and perceived distributional equity, which is not significant in this research.

Moreover, our results reveal that village is significantly related to their perceived recognitional fairness (Beta = 0.118, *p* = 0.014). This can be clearly illustrated by the fact that recognitional fairness perceptions of residents living inside TGPNP are the lowest (Luoyigou village, mean = 3.36), while those from the gateway community are the highest (Yinping village, mean = 3.71). This phenomenon is not complex to comprehend. For one thing, residents of Luoyigou village are more likely to develop negative judgements toward the recognitional indicators of livelihood, legal and traditional rights, as well as land ownership, as they suffer from more strict land use restrictions and more intense human-wildlife conflict, compared to villagers living outside the boundary of TGPNP. For another, the gateway community, Yinping village, has long been supported financially and technologically to develop eco-tourism by the county government and ATA, therefore locals' feelings of recognitional justice are more likely to be positive-going. Similarly, O. Digun-Aweto et al. (2018) [35] found that communities living close to the national park showed more negative attitudes toward conservation, while communities living far away from the national park were not severely impacted by wildlife-caused crop losses and consequently developed more positive perceptions. Apart from this, another factor affecting perceived recognitional equity is household size, which was discovered in our study. Similar results

are noted by Ding Ya (2019) [36] and Liu Yucheng et al. (2018) [37], stating that respondents with larger household size are more likely to be satisfied with ecological compensation and the implementation of programs.

Other demographic features (e.g., gender, annual household income, and age), although they do not pass our regression analysis, have been discussed heatedly in other literature. Georgina G. Gurney et al. (2021) [16] discovered that men are more likely to develop fairness perceptions than women toward merit and equality principles. In addition, Carolina T. Freitas et al. (2020) [22] similarly believed that co-management of fisheries could promote gender equity. However, this phenomenon was not found in TGPNP, where organized co-management activities imposed no apparent gender restrictions on participants. Moreover, a significant positive correlation was detected between annual household income and perceived fairness in our study. This result is consistent with the findings of Nathan J. Bennett et al. (2020) [15], who argued that people with higher relative wealth had more earnings and therefore would have a positive perception of distributional equity. A study by Zhu Ting et al. (2012) [38] further indicated that participation in co-management programs had a significant positive impact on household income. However, Georgina G. Gurney et al. (2021) [16] argued that stakeholders with more material assets are more likely to perceive the distribution of benefits as unfair. Finally, our research also verified the findings by Nathan J. Bennett et al. (2020) [15], in which increasing age is associated with worsening perceptions of recognition, distributive, and integrated equity. This is possibly due to the fact that some co-management activities in TGPNP (e.g., forest rangers and rural tourism skills training) set age restrictions for participants, which lead to the fact that elder villagers with fewer participative opportunities were less likely to develop fairness perceptions.

*4.2. Participative Co-Management Activities and Their Associated Fairness Perceptions*

All participated types of co-management are positively associated with certain dimensions of fairness perceptions. First, the instructive type of co-management, where the government is completely taking control, is significantly associated with perceived distributional equity (Beta = 0.100, $p$ = 0.041), as well as combined social equity (Beta = 0.108, $p$ = 0.021). This is due to the fact that most of these co-management activities which are dominated by ATA help in enhancing local livelihood, such as energy transformation, industrial support, and technology improvement [31]. Those economically supportive activities, in return, are exchanged for conservation obedience of local residents, the enforcement of which can improve local perceptions toward distributional fairness [20].

Second, the consultative type of co-management is recognized as the most widely correlated factor, which is positively associated with all equity dimensions (recognition, Beta = 0.130, $p$ = 0.026; procedure, Beta = 0.283, $p$ = 0.000; distribution, Beta = 0.127, $p$ = 0.016; and combined equity, Beta = 0.243, $p$ = 0.000). This highlights the crucial function of information exchange between communities and ATA staff, if timely and sufficient, it can greatly enhance the identity recognition, participative channels, and opportunities, and promote more equitable benefits sharing of communities. This finding expands the discovery by Catherine Gross (2007) [39] in which access to adequate information is important for procedural fairness, and further detects its effects on recognitional, distributional, and composite justice. By contrast, the agreement co-management type is merely correlated with distributional equity (Beta = 0.104, $p$ = 0.037). This is due to the fact that most of the agreements already signed focus on dealing with economic losses or the redistribution of benefits, such as the agreements of human-wildlife conflict compensation and Chinese beekeeping benefit-sharing. However, only a small proportion of local households have reached agreements with ATA, with the percentages for A7, A8, and A9 as 18.9%, 9.2%, and 24.1%, respectively. This can well explain why participated co-management agreements have no direct influence on combined equity, as the participation scope is not wide enough to exert a comprehensive impact.

Finally, involvement in cooperative co-management activities, such as forest patrolling and enacting conservation rules, has a strong effect on residents' feelings of justice, especially in the procedural, distributional, and combined dimension (Beta = 0.257, 0.300, and 0.295, respectively). It is not difficult to understand its dramatic effect since this type of co-management highly empowers locals. An interesting phenomenon here is that most of the respondents who fairly collaborated with the ATA are local elites, such as village directors, cadres, and rangers, etc. Those elites are extensively exposed to, sufficiently involved in, and fully responsible for those collaborative activities, in order that they are more likely to perceive fair procedures than the ordinary residents, and consequently bestowed with more equitably distributed benefits. This result is consistent with the finding by Haiyun Chen et al. (2012) [40] that members of village councils and co-management committees involved in more projects can enjoy more equitable treatment as a result.

### 4.3. Recommendations for TGPNP

Our results disclose that the relatively low empowering levels and limited numbers of participative activities in co-management, can consequently lower the fairness perceptions toward TGPNP. Most of the respondents in our survey have merely been involved in less than two types of co-management activities, and more specifically, at the lowest instructive empowering level. This dilemma has not been appropriately solved despite the fact that various co-management interventions lasted more than four decades in TGPNP, partially due to the lack of conservation capacity among locals [31]. Another reason for this phenomenon is the scarcity of participative channels, as an interviewee complained: "If the ATA asks me to give suggestions or get involved in conservation affairs, I am very willing to do; but the situation is that they would never ask me". By analyzing the negatively perceived indicators, including the livelihood and land ownership for recognitional equity, participation, FPIC, and decision-making for procedural equity, as well as ecological compensation and human-wildlife conflicts for distributional equity, we can further detect some potential issues faced with TGPNA. It is self-evident that the land grabbing and limitations on traditional livelihoods are common issues facing worldwide protected areas [41–43], and insufficient industrial support, untransparent procedures, inappropriate compensation, and other managerial shortcomings may hinder locals to develop fairer judgements toward the TGPNA.

Based on the aforementioned issues, we believe that facilitating local participation in diversified co-management arrangements can effectively promote more equitably managed protected areas. In this direction, we suggest that the ATA strengthen the publicity of co-management to locals, particularly for the elder and low-income groups, setting more channels for participation, and simultaneously, enhancing conservation awareness and capability of locals. Those countermeasures can improve the participative rates of locals and gradually enhance their empowering levels, after years of attempts and endeavors. Apart from this, the impacts of conservation initiatives on land use and traditional livelihoods need to be addressed urgently [44,45]. For this, we recommend the ATA to facilitate alternative livelihood, such as eco-agriculture and eco-tourism, and provide more job opportunities for locals to get involved in conservation, especially for residents living inside the TGPNP. Moreover, it is imperative to better inform and involve locals in the co-management meetings, planning consultations, and capability-building workshops, in order to set up with fairer participative procedures. Furthermore, we suggest strengthening wildlife monitoring and developing a more equal and reasonable compensation plan to relieve the human-wildlife conflicts and strive for better distributional equity [46].

### 4.4. Future Research

One novel contribution of this research is that it discloses the effects of participative co-management activities on perceptions of recognitional, procedural, distributions, and combined equity. Nevertheless, some limitations remain. First, as the co-management of GPNP is mostly conducted at the instructive level, the research findings can be different

in the contexts of more empowered co-managed regime. Moreover, the survey sampling can spread to a broader range of age groups, especially for the younger generations, since more than half of the current respondents are in their fifties or sixties. Furthermore, additional in-depth interviews can be conducted with local people to capture their deeper understandings toward equity, since this concept can have differentiated meanings to different groups.

Our research finds that the location of villages is a critical element in impacting perceptions of procedural equity. In-depth, we speculate that other spatially related factors (e.g., the accessibility of the residence, the distance from the main road, and entrance of the protected area) might also have essential influence on recognitional and possibly other fairness perceptions. Therefore, we recommend that future researches should focus on this direction to explore the correlation between spatial factors and locals' perceptions of fairness. Moreover, with informal interviews, we realize that multiple stakeholder groups show different perceptions toward various co-management projects. Therefore, we suggest that in-depth interviews and participating observations should be adopted in future studies to compare and contrast fairness perceptions among different stakeholder groups [23].

Furthermore, there is a need to assess the correlation among fairness perceptions, satisfaction degrees, and conservation attitudes, in which a non-linear and complex statistic model might be required. Locals' perception of fairness may affect their satisfaction with and conservation attitudes toward protected areas, and to understand this relationship, it is conducive to achieve the conservation success [47–49].

Finally, both effectiveness and equity are essential, yet different and interdependent concepts in the conservation of protected areas (Woodley et al. 2012 [50]; Schreckenberg et al. 2016 [3]). Some scholars believe that the effectiveness of protection is often achieved without perfect social equity (Klein et al. 2015 [7]; Dawson et al. 2017 [51]), indicating that they may not be simply positively correlated. Therefore, pursuing extreme equity in protected areas is encouraged, but it is more worthwhile to explore the extent of equity that can achieve maximum efficiency. Community-based co-management, in the context of protected areas, serves as a crucial means to balance both effective and equitable management (Persha and Andersson 2014 [52]) [20]. Therefore, a greater focus on analyzing the relationship and trade-off between social equity and conservation effectiveness of co-managed protected areas can produce thought-provoking findings, and better conduct management effectiveness evaluation with consideration of social equity.

## 5. Conclusions

For the broader well-being of the local people and stakeholders, the equity issues in marine and terrestrial protected areas are receiving increasing attention globally. However, despite the co-management approach being widely promoted worldwide for the better governance of protected areas, little attention has been paid to its effects on fairness perceptions. This paper builds on the considerable work in social equity issues and the empowering levels of co-management to further explore the correlation between participated co-management arrangements and perceived social equity of locals, from recognitional, procedural, and distributional dimensions. The main conclusions are summarized below: (1) There is a distinct variability in fairness perceptions toward TGPNP, with the recognitional equity as positive, procedural, and distributional negative, and the combined equity as neutral. (2) The participated co-management activities and reflected empowering levels of locals are rather limited, with most of the respondents remaining at the instructive level. The number of participants declines with the increase in empowering levels. (3) Participation in diversified co-management activities is revealed to be influential on locals' fairness perceptions. While the consultative type of co-management is recognized as the most widely correlated factor, the cooperation is found to have the strongest impact. By contrast, the impact scope of instruction and agreement types of co-management are pretty narrow, mostly in the distributional dimension. (4) With regards to the demographic features, education is found to be positively related to all equity perceptions, while village

is significantly merely for recognitional equity. These findings indicate that the more locals are involved in co-management activities, the fairer they are likely to perceive the protected areas. This points to the need for more empowered and widely involved co-management plans to improve social equity judgement in TGPNP. Furthermore, regarding specific affairs (e.g., ecological compensation and human-wildlife conflicts) or communities of different locations (e.g., communities inside or outside protected areas), tailored countermeasures should be taken for better consideration of social equity issues.

This paper highlights the critical importance of exploring social equity in the co-management arrangements of nationally designated protected areas. To promote the achievement of the fairness goals and conservation goals for a broader population, we encourage the global conservation community to conduct more discussions that combine social equity and co-management issues, which can consequently produce more co-management plans, principles or instructions with equity consideration.

**Supplementary Materials:** The following supporting information can be downloaded at: https://www.mdpi.com/article/10.3390/land11101624/s1. Table S1: Survey questions related to the demographics and characteristics of local residents; Table S2: Survey questions related to participated types of co-management local residents; Table S3: Survey questions and responses related to recognitional, procedural, and distributional equity; Table S4: Descriptive summary of survey sample including demographics and characteristics of local residents; Table S5: Descriptive summary of survey sample including the number and percentage of local residents that participated in co-management types and activities; Table S6: Descriptive summary of survey sample including the quantity of co-management participation of local residents; Table S7: Descriptive summary of responses to all individual perception indicators; Table S8: Correlation analysis between demographic characteristics and perceived fairness; Table S9: Correlation analysis of co-management participation type and fairness perception; Model S1: A regression model of recognitional equity and relevant independent variables; Model S2: A regression model of procedural equity and relevant independent variables; Model S3: A regression model of distributional equity and relevant independent variables; Model S4: A regression model of combined social equity and relevant independent variables.

**Author Contributions:** Conceptualization, Y.Z. (Yin Zhang) and Y.Z. (Yuqi Zhang); methodology, Q.C. and Y.Z. (Yin Zhang); software, Q.C.; formal analysis, Q.C.; investigation, Q.C., Y.Z. (Yuqi Zhang) and Y.Z. (Yin Zhang); data curation, Q.C.; writing—original draft preparation, Q.C., Y.Z. (Yuqi Zhang) and Y.Z. (Yin Zhang); writing—review and editing, Q.C., Y.Z. (Yuqi Zhang), Y.Z. (Yin Zhang) and M.K.; visualization, Q.C.; supervision, Y.Z. (Yin Zhang); project administration, Y.Z. (Yin Zhang); funding acquisition, Y.Z. (Yin Zhang). All authors have read and agreed to the published version of the manuscript.

**Funding:** This research was funded by the National Natural Science Youth Program, grant number 52108040, and the China Postdoctoral Science Foundation, grant number 2021M700574.

**Data Availability Statement:** Not applicable.

**Acknowledgments:** The authors would like to thank all the anonymous reviewers and editors who contributed their time and knowledge to this study. The authors also thank Peng L., Liang Y.L., Wang L.C., Li Q.Y., Shen B.S., Chen J.X. and Zuo C.L. who were involved in the tough field survey in torrid summer. Thanks to Zhang Z.Q. and Li K. for their precious support for our survey. And thanks to all staffs of ATA who provided us with documentation and materials, or helped with our interviews and questionare distribution. We also own special thanks to Dan Brockington, an interview with whom initially inspired the corresponding author to develop the preliminary idea of this research. We authors would like to thank Huang J.L. sincerely for his strong statistical support and guidance of our study.

**Conflicts of Interest:** The authors declare no conflict of interest.

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
