# Peer review of "Examining Social Equity in the Co-Management of Terrestrial Protected Areas: Perceived Fairness of Local Communities in Giant Panda National Park, China"

_land, doi:10.3390/land11101624_

Round 1
Reviewer 1 Report
Dear authors,
your paper deals with a very interesting issue since assessing how local people living in the surrounding of PAs perceive social equity is essential to ensure a more fair PA management. The paper is well structured, methods and analysis are consistent and results are interesting. However, there are several methodological shortfalls that I suggest improving (see my detailed comments in pdf attached) and the following considerations that you could reflect about.
My first suggestion is that you should better describe your variables and their role in shaping social equity. This is intrinsically linked to your method since once you build a (univariate or multivariate) model for predicting factors shaping perceptions and attitudes you are making hypothesis. In other words, what are your hypotheses? And why you choose some variables instead than others? This should be clearer in your work. For example, one can suppose that gender has a strong role in shaping social equity considering gender inequality etc. Have you considered this hypothesis? And if so, why you do not include and/or test the gender variable in your analysis?
My second suggestion is that your discussion could be improved. I think you should include some considerations about what one would expect from the results (for example, based on your hypotheses!). You are describing and interpreting what you have found (interesting!!!), but what about the results you were expecting and they did not outcome. For example, why gender or ‘migration status’ are not significant in shaping the perception of social equity in your context? Moreover, your results emphasise the positive role of formal education in perceiving social equity, while age seem to be not influent. However, one would suppose that older people are less educated and that there could be a strong correlation between age and education. Have you considered this aspect? If so, what are your results showing concerning this likely relationship?

Reviewer 2 Report
Social equity is one of the key topics for protected area governance and management worldwide in the recent two decades. This manuscript uses a newly established national park in China to examine the items and degree of social equity for the locals the national park has achieved. This is a brilliant manuscript, structuralized, comprehensive, and reasonable. I do learn a lot from the reading. However, there is always something we could improve regarding the readability and quality. Firstly, there is something that might be good to add. For example, some of the community-based efforts have been implemented long before the establishment of the national park. Thus, it may be better to mention briefly those in situ institutions before the national park. Also usually there are community-based (or grassroots) organizations, such as clans, family groups, religious groups, etc., and other social organizations (institutions, traditional or modern) in the local communities, who usually play a key role in the general life locally. I wonder if there could be some more words to enrich the background information. On line 150, it mentioned 12 co-management activities were chosen from 15 ones with ATA staff. It might be great to mention the reason(s) and have the list of all 15 items in supplementary materials. Secondly, I would encourage the authors to describe the limitations of the research or self-review the methodology and materials. Thirdly, I cannot find the reference (not [27]) for the five types of co-management arrangements on line 76. Please do add it. Fourthly, it might be good to have someone for English editing, like checking the terminology and characters, such as participated, participative, and participatory. It is better to adopt the same terminology throughout the manuscript. For example, as household size is first adopted, it might be not necessary to change to family size. Fifthly, maybe the authors can consider taking table S4 to replace table 4 for only one item different. For the migrant workers of the survey items in table 4, some more information might be good, i.e. (no./family). Sixthly, there is still some potential for analysis and discussions, even future research. For example, management effectiveness evaluation and green list, which the authors could consider.
